behaviour, neuroscience

mosquito, behaviour, *Aedes aegypti*, larvae, chemotaxis

**Author for correspondence:**
Jeffrey A. Riffell
e-mail: jriffell@uw.edu

# Computational and experimental insights into the chemosensory navigation of *Aedes aegypti* mosquito larvae

Eleanor K. Lutz[1], Tjinder S. Grewal[2] and Jeffrey A. Riffell[1]

[1]Department of Biology, University of Washington, Box 351800, Seattle, WA 98195, USA
[2]Department of Biochemistry, University of Washington, Box 357350, Seattle, WA 98195, USA

EKL, 0000-0002-2434-2254; JAR, 0000-0002-7645-5779

Mosquitoes are prolific disease vectors that affect public health around the world. Although many studies have investigated search strategies used by host-seeking adult mosquitoes, little is known about larval search behaviour. Larval behaviour affects adult body size and fecundity, and thus the capacity of individual mosquitoes to find hosts and transmit disease. Understanding vector survival at all life stages is crucial for improving disease control. In this study, we use experimental and computational methods to investigate the chemical ecology and search behaviour of *Aedes aegypti* mosquito larvae. We first show that larvae do not respond to several olfactory cues used by adult *Ae. aegypti* to assess larval habitat quality, but perceive microbial RNA as a potent foraging attractant. Second, we demonstrate that *Ae. aegypti* larvae use chemokinesis, an unusual search strategy, to navigate chemical gradients. Finally, we use computational modelling to demonstrate that larvae respond to starvation pressure by optimizing exploration behaviour—possibly critical for exploiting limited larval habitat types. Our results identify key characteristics of foraging behaviour in an important disease vector mosquito. In addition to implications for better understanding and control of disease vectors, this work establishes mosquito larvae as a tractable model for chemosensory behaviour and navigation.

## 1. Introduction

The mosquito *Aedes aegypti* is a global vector of diseases such as Dengue, Zika and Chikungunya [1]. This synanthropic mosquito is evolutionarily adapted to human dwellings, with some populations breeding exclusively indoors [2,3]. The urban microhabitat features unique climatic regimes, photoperiod and resource availability. In response to these selective pressures, successful synanthropic animals including cockroaches [4], rats [5] and crows [6] exhibit many behaviours absent in non-urbanized sibling species. Understanding these behaviours is of major importance to public health. Throughout human history, synanthropic disease vectors have caused devastating pandemics like the Black Death, which killed an estimated 30–40% of the western European population [7,8]. Like rats or cockroaches, adult *Ae. aegypti* mosquitoes exhibit many behavioural adaptations to human microhabitats [2,9]. However, comparatively little is known about larval adaptations. The larval environment directly affects adult body size [10,11], fecundity [10] and biting persistence [12], and understanding vector survival at all life stages is crucial for improving disease control [13]. Despite growing interest [14–16], it remains an open question how environmental stimuli affect larval behaviour to regulate these responses and processes.

In addition to the above public health implications, the behaviour of synanthropic mosquito larvae is fascinating from a theoretical search strategy perspective. *Aedes aegypti* larvae are aquatic detritivores that live in constrained environments such as vases and tin cans [11]. In such limited environments, do

larvae exhibit a chemotactic search strategy (in which animals change their direction of motion in response to a chemical stimuli), or do they use a chemokinetic response (in which animals change a non-directional component of motion, such as speed or turn frequency, in response to a chemical stimuli) [17]? Mechanistic understanding of larval foraging behaviour may provide insight into chemosensory systems controlling the behaviour as well as the evolutionary adaptations for these systems in synanthropic environments.

In this work, we investigate larval *Ae. aegypti* behavior from a chemical ecological and search theory perspective. First, we explore chemosensory cues involved in larval foraging. Although many olfactory cues are used by adult females to select oviposition sites [18], it is unclear if larvae and adults use the same chemicals to assess larval habitat quality. Second, we consider larval search behaviour in spatially restricted environments using empirical data and computational modelling. Our work identifies the lack of chemotaxis in foraging *Ae. aegypti* larvae—an example of how environmental restrictions may drive the evolution of animal behaviour. We further identify microbial RNA as a potent and unusual larval foraging attractant. Together, our results identify *Ae. aegypti* larvae as a tractable model in biological search theory, and highlight the importance of investigating synanthropic disease vectors at all life-history stages.

## 2. Results

### (a) Effects of sex, physiological state and circadian timing on larval physiology

Behavioural experiments in insects have demonstrated the importance of circadian timing, starvation and age [19]. However, little is known about the effects of these variables on *Ae. aegypti* larvae. To better understand the effects of nutritional state and sex on our study organism, we used machine vision to track individual 4th instar *Ae. aegypti* larvae in a custom arena before each experiment (figure 1a). For both fed and starved animals, female larvae were larger than males (fed larvae: $n = 135$ female, 153 male, $p < 0.0001$, effect size = 0.53 mm; starved larvae: $n = 89$ female, 122 male, $p < 0.01$, effect size = 0.26 mm, electronic supplementary material, figure S1A). Starved larvae were also smaller than fed animals for both females ($p < 0.0001$, effect size = 0.51 mm) and males ($p < 0.01$, effect size = 0.23 mm, electronic supplementary material, figure S1A). Because adult *Ae. aegypti* exhibit crepuscular activity [11], we also investigated the effects of circadian timing on larval behaviour. We found no effects of circadian timing on larval movement speed ($p = 0.40$), time spent moving ($p = 0.41$), or time spent next to the arena walls ($p = 0.55$). These observations support previous findings that mosquito larvae, unlike adults, exhibit little behavioural variation during the day [21,22].

### (b) Quantifying the chemosensory environment in naturalistic larval habitat sizes

Previous research has shown that other species of mosquito larvae detect many different chemosensory stimuli [23]. In *Ae. aegypti*, it is unclear what chemical cues, if any, larvae use to navigate their environment. Nevertheless, chemosensory cues may be essential in avoiding predation

or foraging efficiently. Using our arena and machine vision methods, we investigated larval preference for eight putatively attractive and aversive sets of stimuli. First, we experimentally verified the chemical diffusion in the arena and found that larval movement significantly increased the diffusion of stimuli within the arena ($p < 0.0001$; electronic supplementary material, figure S2). We next created a chemical diffusion map for analysing stimuli preference using only experiments containing larvae (figure 1b; electronic supplementary material, figure S2A-D). For chemosensory stimuli, we used predicted attractive stimuli including a 0.5% mixture of food (Hikari Tropic First Bites fish food) suspended in water, as well as food extract filtered through a 0.2 µm filter to remove solid particulates. Quinine was used as a putative aversive stimulus (a bitter tastant aversive to many insects including *Drosophila melanogaster* and *Apis mellifera* [24,25]). We also tested indole and o-cresol, two microbial compounds that attract adult mosquitoes for oviposition [26]. Finally, we tested the response of larvae to RNA, glucose and a mixture of nine amino acids required for *Ae. aegypti* larval growth. All three components are essential for *Ae. aegypti* survival [27], and RNA polynucleotides serve as attractants or essential nutrients for larvae of other mosquito species [28–31]. Moreover, dissolved RNA is released at high levels (µg $l^{-1}$ $h^{-1}$) from growing populations of microbes in freshwater habitats [32], and could provide valuable foraging information to omnivores such as *Ae. aegypti*. By contrast, other isolated macronutrients such as salts, sugars and amino acids elicit little to no attraction in other larval mosquito species [33].

### (c) Physiological feeding state affects larval attraction towards ecologically relevant odours

For each of these eight sets of stimuli, in addition to water, we compared the stimulus preference of larvae before and after stimulus addition (figures 1c, 2a; electronic supplementary material, figures S3–S5). Preference was defined as the median concentration chosen by the larvae throughout the 15 min experiment, normalized to behaviour during the previous 15 min acclimation phase. Starved larvae were attracted to food ($n = 32$, $p < 0.0001$) and spent significantly less time near the aversive cue quinine ($n = 19$, $p < 0.0001$). Food extract filtered through a 0.2 µm filter remained attractive ($n = 19$, $p = 0.004$), suggesting that larvae use small, waterborne chemical cues to forage. To further investigate these foraging cues, we next examined responses to microbial RNA, glucose and an amino acid mixture. We found that RNA was significantly attractive ($n = 18$, $p = 0.049$), while glucose ($n = 20$, $p = 1$) and the amino acid mixture ($n = 23$, $p = 1$) were not. Addition of water—a negative control for mechanical disturbance–had no impact on larval positional preference ($n = 16$, $p = 1$). Although we expected indole and o-cresol, which are attractive to adult *Ae. aegypti*, to elicit attraction from larvae, neither odorant elicited a change in behaviour from the acclimation phase (indole: $n = 20$, $p = 1$; o-cresol: $n = 25$, $p = 1$). Indole tested at a higher concentration (10 mM) also had no effect ($n = 19$, $p = 0.31$). Together, these results suggest that larvae and adults may not necessarily rely on similar cues to assess larval habitat quality.

The physiological feeding state of an adult mosquito has a strong impact on subsequent behavioural preferences [34], but it remains unknown how feeding status influences responses to

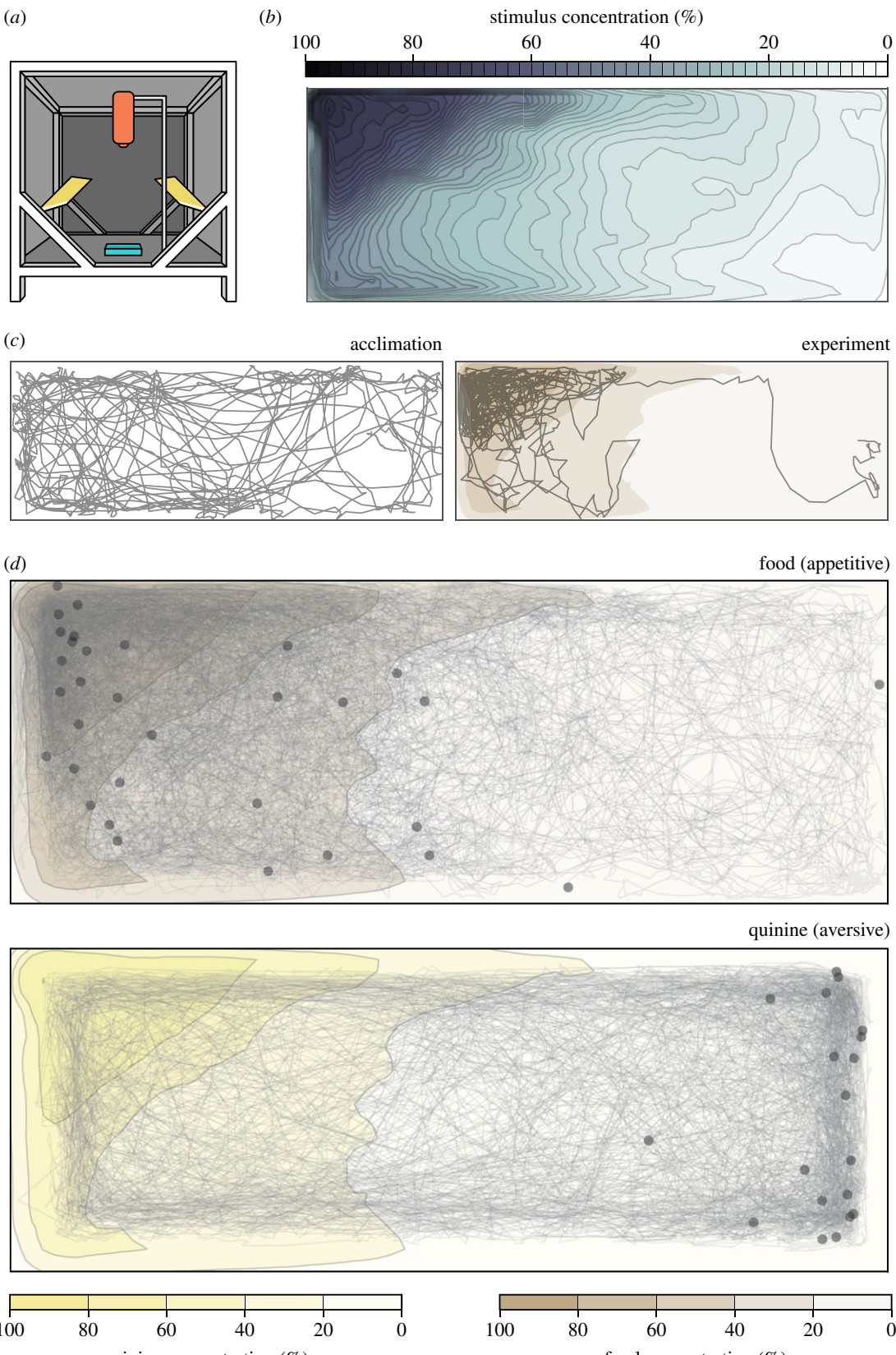

**Figure 1.** Quantifying the chemosensory environment in naturalistic larval habitat sizes. (*a*) Diagram of experimental conditions, adapted from [20], including a Basler Scout Machine Vision GigE camera (orange), infrared lighting (yellow) and a behaviour arena (blue). (*b*) Chemosensory diffusion map of the behaviour arena at the end of the 15 min experiment. (*c*) Example of an individual larval trajectory during the 15 min acclimation phase (left). Trajectory of same individual during the 15 min experiment phase, responding to food added to the left side of the arena (right). (*d*) Trajectory of all starved animals presented with food (top) or quinine (bottom). Although trajectories are shown aggregated into one image, all animals were tested individually. Scatter points show the position of each animal at the end of the experiment and colour overlays show the chemosensory diffusion map at the end of the 15 min experiment. (Online version in colour.)

chemosensory stimuli in larvae. We thus fed larvae ad libitum fish food before testing their responses to each of the eight stimuli and a water control (figure 2*b*). Fed larvae showed

no significant attraction to food ($n = 57$, $p = 1$), food extract ($n = 19$, $p = 1$) and RNA ($n = 20$, $p = 1$), supporting the prediction that microbial RNA functions as an attractant in the

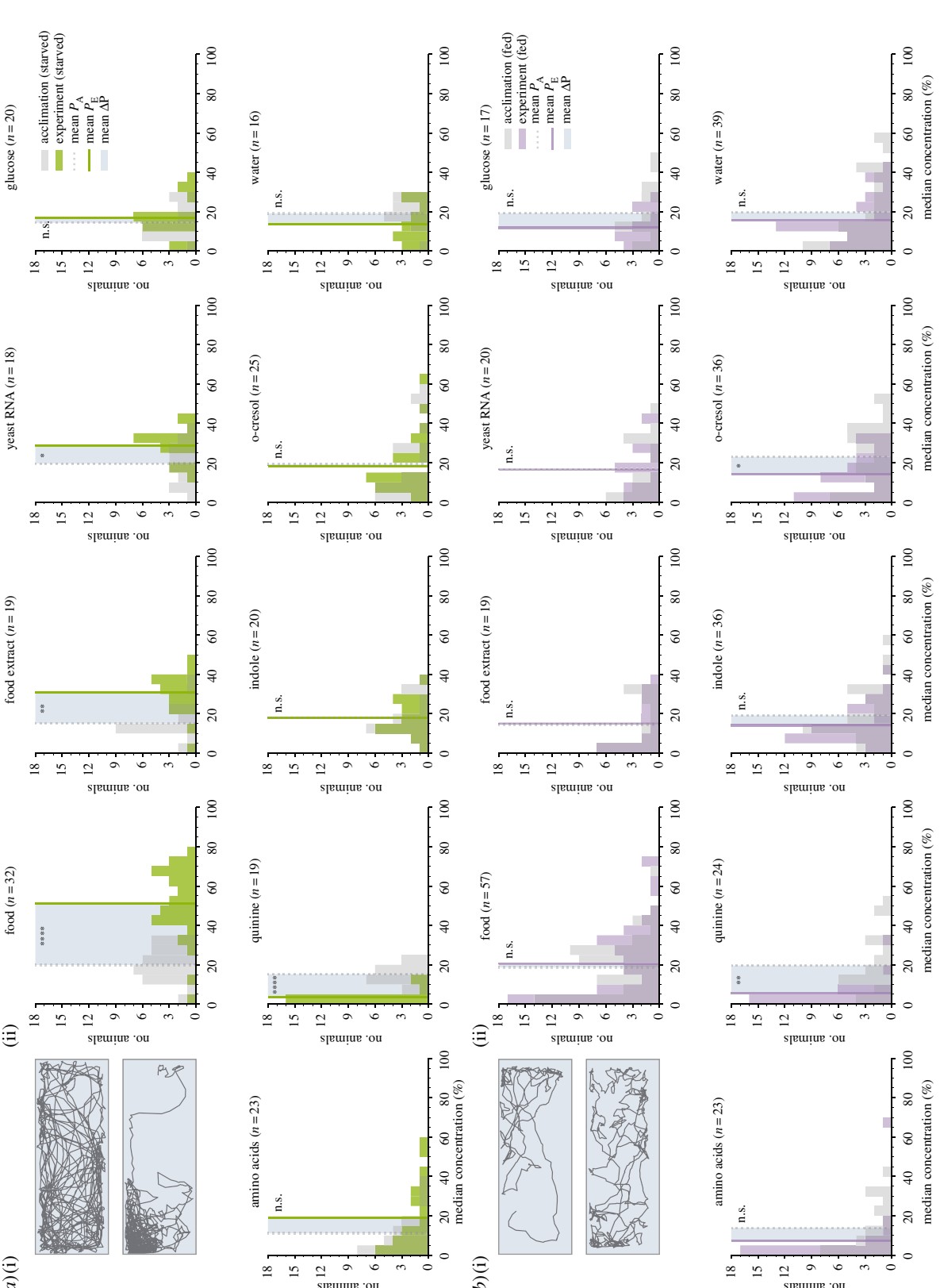

**Figure 2.** Physiological feeding state affects larval attraction towards ecologically relevant odours. (*a*(i)) Example trajectory of a starved larva during the acclimation (top) and the experiment phase (below), responding to food introduced to the top left. (*a*(ii)) Distribution of larvae during the acclimation phase (grey) and experiment phase (green), median concentration. The shaded box visualizes the mean ΔP across all individuals. Note that owing to the unequal distribution of high and low concentration areas in the behaviour arena, animals naturally appear to distribute near lower concentrations when no stimulus is present. (*b*(i)) Example trajectory of a fed larva during the acclimation (top) and experiment phase (below), responding to food introduced to the top left. (*b*(ii)) Distribution of fed larval preference during the acclimation (grey) and experiment phase (purple). In ((*a*)ii) and ((*b*)ii), asterisks denote the significance level of paired-sample Welch's *t*-tests comparing acclimation P and experiment P (n.s., not significant). *n* values reported next to each stimulus describe the number of animals in the treatment. (Online version in colour.)

**Table 1.** Comparing larval exploration behaviour to canonical animal search strategy models. (Four different chemosensory search strategies are listed (central columns) along with the expected observable behaviour metrics for each strategy (left column). By comparing the experimental observations (right column) with the expected results, we determined that *Ae. aegypti* larval chemosensory navigation is best explained by a chemokinesis search strategy model. Cells in bold type indicate expected experimental results (anosmic, chemotaxis, klinotaxis, and chemokinesis) or statistically significant observed results (far right column).)

| | potential chemosensory search strategies | | | | |
| --- | --- | --- | --- | --- | --- |
| | anosmic | chemotaxis | klinokinesis | chemokinesis | experiment observations |
| stimulus preference ΔP | no | **yes** | **yes** | **yes** | **yes ($p < 0.0001$)** |
| directional preference ΔDP | no | **yes** | no | no | no ($p = 0.98$) |
| Δ concentration speed ΔDS | no | no | no | no | no ($p = 1$) |
| concentration speed ΔCS | no | no | no | **yes** | **yes ($p < 0.0001$)** |
| Δ concentration turns ΔDTI | no | **yes** | no | no | no ($p = 1$) |
| concentration turns ΔCTI | no | no | **yes** | no | no ($p = 1$) |

context of foraging. Nonetheless, fed larvae still exhibited aversive responses to quinine ($n = 24$, $p = 0.003$), demonstrating that the lack of response to foraging cues is not because of a global reduction in chemosensory behaviour. Similar to starved larvae, fed animals showed no preference for the water control ($n = 39$, $p = 1$), indole (100 μM: $n = 36$, $p = 0.98$; 10 mM: $n = 17$, $p = 1$), glucose ($n = 17$, $p = 1$) or the amino acid mixture ($n = 23$, $p = 1$). Fed larvae exhibited significant aversion to o-cresol ($n = 36$, $p = 0.026$).

## (d) A chemokinesis navigation strategy is most consistent with larval aggregation towards cues investigated in this study

Next, we investigated the behavioural mechanism by which *Ae. aegypti* larvae locate sources of odour, because such information could provide insight into the chemosensory pathways that mediate the behaviours. We hypothesized that larval aggregation near attractive cues such as food is mediated by chemotaxis—a common form of directed motion observed in many animals and microbes [35–37]. In chemo-klino-taxis (hereafter chemotaxis), animals exhibit directed motion with respect to a chemical gradient. Alternatively, larvae may exhibit chemo-ortho-kinesis (hereafter chemokinesis)—a process in which animals respond to local conditions by regulating speed rather than direction—or chemo-klino-kinesis (hereafter klinokinesis)—in which animals respond to local conditions by regulating turning frequency. Finally, larvae may be unable to detect chemosensory stimuli, and thus exhibit purely random behaviour (hereafter anosmic). To differentiate between these strategies, we quantified six observable metrics used to characterize navigation behaviour (table 1). By breaking down larval trajectories into several different components (figure 3a,b) and identifying which variables correlate with stimulus preference (figure 3c,d), we can infer which search strategy best explains larval behaviour.

Surprisingly, we found no evidence for chemotaxis near attractive or aversive chemicals. Starved larvae did not exhibit kinematic changes characteristic of chemotaxis, such as directional preference (ΔDP, $p = 0.98$; figure 3b(i); electronic supplementary material, figure S6A). Furthermore, larvae could not increase odour localization efficiency above random chance: discovery time for all cues was comparable across treatments (ΔD, $p = 1$; electronic supplementary

material, figure S6B). Larvae also did not perform klinokinesis: turning frequency was unaffected by either the instantaneous concentration the larvae experienced (ΔCTI, $p = 1$; figure 3b(ii); electronic supplementary material, figure S6C) or change in concentration (ΔDTI, $p = 1$; electronic supplementary material, figure S6D). Instead, we found that larval activity was most consistent with chemokinesis for the eight cues tested in these experiments. Larvae altered movement speed when experiencing high local stimuli conditions (ΔCS, $p < 0.0001$; figure 3d) but not when moving up or down the concentration map (ΔDS, $p = 1$; electronic supplementary material, figure S6E). When grouped into aversive, attractive and neutral chemosensory cues, the correlation between preference (ΔP) and chemokinetic response (ΔCS) similarly separated into three clusters (electronic supplementary material, figure S7). We did not observe a strong linear relationship in our dataset, perhaps because the majority of cues tested did not elicit a strong behavioural preference.

## (e) Starved *Aedes aegypti* optimize exploration behaviour to increase the probability of finding food

Many organisms change their speed or activity rate when starved [38], and we predicted that starved *Ae. aegypti* may also alter their exploration behaviour to increase the probability of discovering food. Experimental observations showed evidence for starvation-mediated behaviour changes—starved animals spent more time exploring ($p < 0.0001$; figure 4a) and spent less time near walls and corners ($p < 0.0001$; figure 4b). We were interested in understanding whether or not these behavioural changes might be adaptive in ecologically relevant container sizes. We thus created two chemokinesis foraging models using empirical data from fed and starved animals ($n = 248$ fed larvae during the acclimation phase: $n = 445\,925$ trajectory data points; $n = 168$ starved larvae during the acclimation phase: $n = 302\,096$ trajectory data points). This computational model explored circular arenas of various ecologically relevant diameter sizes 5–20 cm in diameter (electronic supplementary material, table S1) by randomly sampling instantaneous speed and turn angle from experimental data (figure 4c). Individual simulations using this model were tasked with finding a food source at the centre of one of these arenas (figure 4d), starting from a randomized location. Similar to the trajectories of starved larvae (figure 2d), our

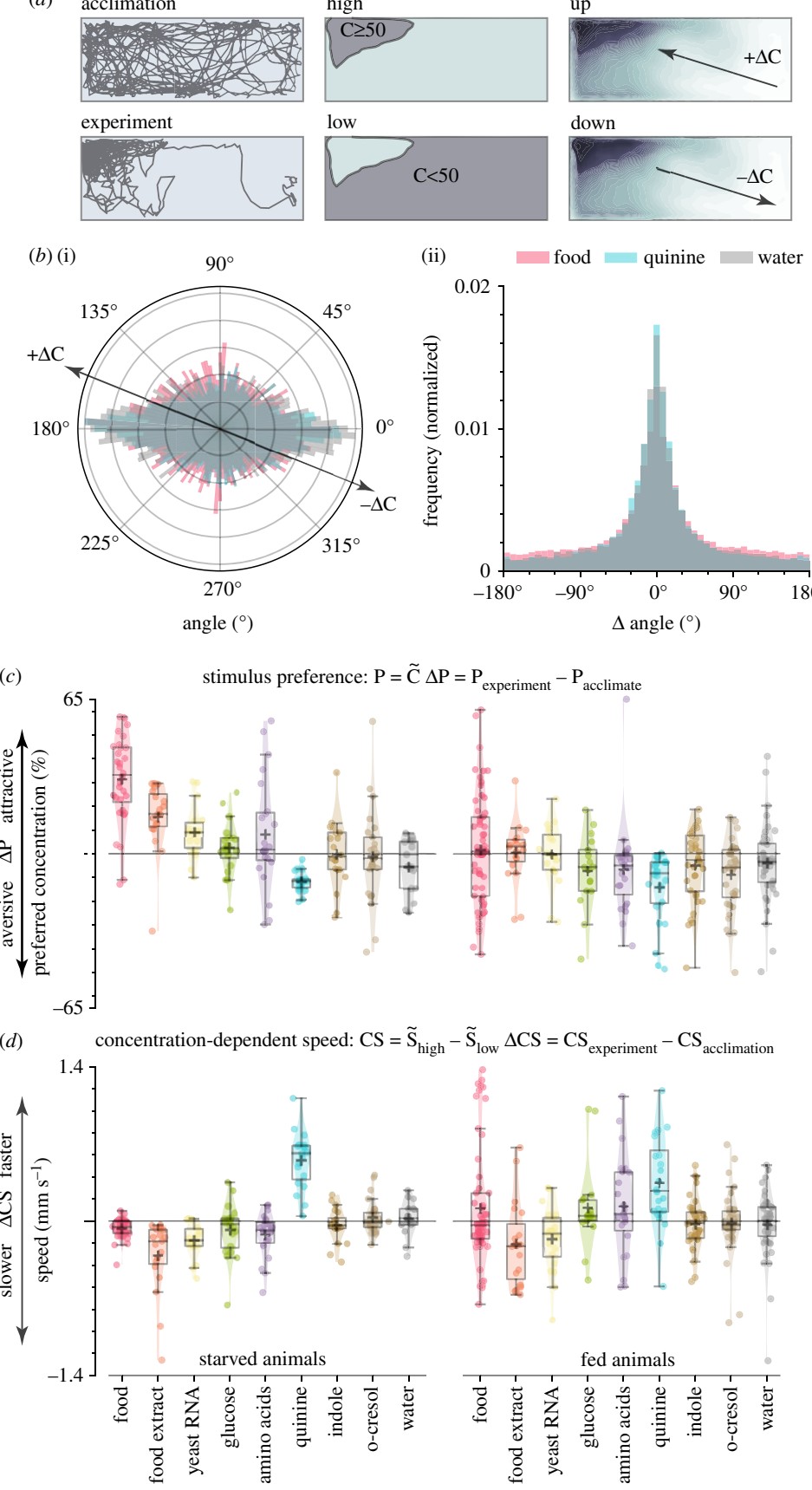

**Figure 3.** Larval exploration behaviour is best explained by a chemokinesis search model. (*a*) Diagram of behavioural quantifications. Larvae were observed during a 15 min acclimation period in clean water, followed by a 15 min experiment in the presence of the stimulus. The arena was divided into an area of high ($\geq$ 50%) and low concentration (< 50%). Larvae could move in a direction that increased local concentration ($+\Delta C$) or decreased local concentration ($-\Delta C$). (*b*)(i) Orientation of animals in the arena throughout the experiment. Larvae did not exhibit directional movement in response to appetitive or aversive stimuli. Note that larvae spend more time moving horizontally (0°, 180°) because the rectangular arena is longer in the horizontal direction. (*b*)(ii) Larvae did not change frequency of turns ($\Delta$ angle) in response to appetitive or aversive stimuli. (*c*) Box plots for the population median $\pm$ 1 quartile, population mean (+ marker) and response for each individual (dots) for larval preference ($\Delta P$). A horizontal line at 0 represents no change in behaviour following stimulus addition. (*d*) As in (*c*), except for stimulus-dependent changes in concentration-dependent speed ($\Delta CS$). (Online version in colour.)

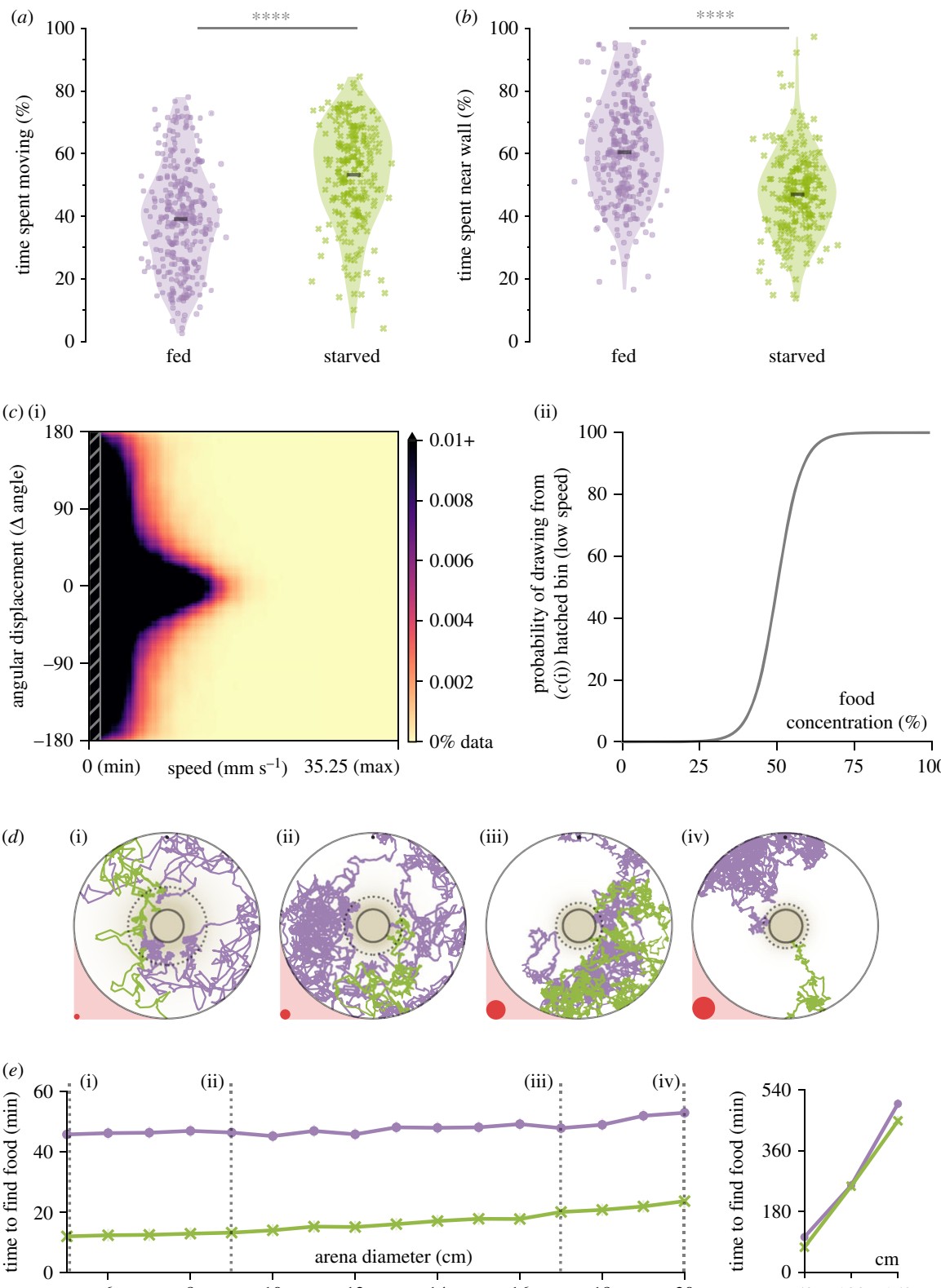

**Figure 4.** Starved *Ae. aegypti* optimize exploration behaviour to increase the probability of finding food. (*a*) Starved larvae spend more time exploring the arena than fed larvae. (*b*) Starved larvae spend less time within one body length of the arena wall. (*a,b*) Violin plot. Dots represent each individual, and black bar is the mean across all individuals (*n* > 168 per treatment); asterisks denote *p* < 0.0001 (Welch's *t*-test). (*c*) We developed a computational model to approximate the chemokinetic behaviour observed in experimental data. (*c*(i)) Probability density function of the relationship between movement speed and instantaneous angle for starved animals. The shaded grey rectangle to the left visualizes the area encompassing half of the available data. (*c*(ii)) Trajectories were constructed by sampling values from the shaded rectangle when larvae were in areas of high food concentration, and sampling values from outside the shaded rectangle when in areas of low food concentration. The probability function for drawing from the two distributions was smoothed to avoid threshold artefacts. (*d*) Simulated trajectories of fed (purple) and starved (green) larvae foraging in ecologically relevant arena sizes. Relative size of each arena is visualized as red circles. In this figure, larvae began at the top centre (fed) or the bottom centre (starved). However, in actual simulations starting location was randomized for each individual. (*d*(i)) 5 cm, (*d*(ii)) 9 cm, (*d*(iii)) 17 cm and (*d*(iv)) 20 cm simulated arena diameters. In all cases, the solid black circle outlines the food target goal, and the dashed circle represents the boundary of 50% food concentration. (*e*) Simulated chemokinetic larvae using empirical data from starved animals (green, X markers) found the food source consistently faster than the same model using data from fed animals (purple, dots). Mean of 1000 simulations ± s.e. Dashed grey lines correspond to ecologically relevant habitat sizes described in the electronic supplementary material, table S1 and in (*d*). (Online version in colour.)

simulated trajectories exhibited tortuous paths that ultimately encountered the food patch. Nonetheless, the chemokinesis model using empirical data from starved animals discovered the food source more than 20 min faster than fed animals across all habitat sizes (figure 4e), supporting our hypothesis that starvation-mediated changes in larval behaviour increase the probability of finding food in larval environments. Moreover, simulated starved larvae could find the food source in under 25 min across these smaller environment sizes (figure 4e). Given that Ae. aegypti larvae can survive up to a week without food [11], our results suggest that a chemokinetic search strategy is sufficient to successfully forage in diverse and realistic larval habitats. Although our simulation assumptions are less suitable for understanding larger breeding sites[1], we further simulated habitats 50, 100 and 150 cm in diameter for comparison. We found that larvae still discovered the food source in several hours (fed simulations: 1.7, 4.3 and 8.3 h; starved simulations: 1.2, 4.2 and 7.5 h for 50, 100 and 150 cm arenas, figure 4e). Finally, the slope for starved animals in smaller habitats was about twice that of fed animals (starved: $45.3 \text{ s cm}^{-1}$; fed: $22.9 \text{ s cm}^{-1}$), suggesting that the benefit of behavioural modification in starved animals is more pronounced in smaller arena sizes (slope of difference between fed—starved simulations: $-22.4 \text{ s cm}^{-1}$).

## 3. Discussion

In this study, we quantify essential characteristics of Ae. aegypti larval behaviour that are crucial for the development of future studies. Furthermore, we identify previously unknown behaviours that highlight the unique evolutionary history and developmental biology of these disease vector mosquitoes. First, we show that larvae perceive microbial RNA as a foraging attractant, but do not respond to several olfactory cues that attract adult Ae. aegypti for oviposition. Second, we demonstrate that Ae. aegypti larvae use chemokinesis, rather than chemotaxis, to navigate with respect to chemical sources. Finally, we use experimental observations and computational analyses to demonstrate that larvae respond to starvation pressure by changing their behaviour to increase the probability of finding food sources in realistic habitat sizes.

Although adult Ae. aegypti feeding is regulated by ATP perception [39], we are unaware of other work demonstrating perception of nucleotides or nucleic acids such as RNA in Ae. aegypti larvae. In our state-dependent preference experiments, we investigate the ecological basis of larval RNA attraction, and propose that RNA may function as one of the foraging indicators in the larval environment. However, 44 different nutrients are required for Ae. aegypti larval survival [27], and the attractiveness of other potential phagostimulants including vitamins and carbohydrates have not been tested with the sensitivity of our experimental methods. In addition, the concentration and relative composition of phagostimulants may have complex effects on larval preference, and these combinatorial effects were not examined in this study. In a natural environment Ae. aegypti larvae probably rely on a combination of stimuli to locate food sources. Nevertheless, an earlier study demonstrated that olfactory receptor deficient (orco −/−) Ae. aegypti larvae showed no defects in attraction to food [20]. Taken together, our results support the hypothesis that sensory information gained from gustatory or ionotropic receptors may be more integral to larval chemosensation than olfactory receptors. Furthermore, larval attraction to RNA suggests that the importance of nucleotide phagostimulation is preserved throughout a mosquito's life cycle, from larval foraging to adult blood engorgement and oviposition.

Our study also raises a number of comparative questions that could be addressed in future research. For instance, is chemokinesis in mosquito larvae associated with human association and man-made containers? Future studies could compare chemotactic ability in other spatially constrained mosquitoes, such as Toxorhynchites (which inhabit tree holes) or Aedes albopictus (another container-breeding mosquito) [40], to species that oviposit in larger bodies of water such as Aedes togoi (marine rock pools) or opportunistic species such as Culex nigripalpus that oviposit in a wide range of habitat sizes [40,41]. Additionally, computational modelling of fluid dynamics and larval movement may help determine whether chemotaxis is physiologically and physico-chemically challenging in small, man-made environments. Owing to the diffusive environment in the small containers, where shallow gradients dominate and turbulence is lacking, the change in time or space of the chemical signal may be too small for the larvae to detect. This is particularly relevant considering our results showing that larval movement significantly modifies the stimulus gradient [42].

Synanthropic mosquitoes are increasingly important to global health as urbanization progresses: currently, over half of all humans live in urban environments, and this proportion is only expected to increase [43]. Adaptations that facilitate human cohabitation, like specialized larval foraging strategies, are vital to our understanding of mosquito behaviour and success as a disease vector [9].

## 4. Material and methods

Details on the insects, selection of preparation of odorants and statistical analyses, can be found in the electronic supplementary material.

### (a) Behaviour arena and experiment

We previously developed a paradigm to investigate chemosensory preference in larval Ae. aegypti [20]. In this study, we expanded our protocol by mapping the chemosensory environment in our arena using fluorescein dye. Importantly, because larval swimming activity increases chemical movement within the arena, we mapped the dye distribution from experiments containing an actively swimming larva. Fluorescein dye (100 µl) was added to a white arena of the same material and dimensions, each containing one Ae. aegypti larva. Dye colour was converted to concentration values using a standardization dataset of 13 reference concentrations (electronic supplementary material, figure S2C). Dye diffusion through time was quantified by the mean of all values in each 1 mm² area, linearly interpolated throughout time (n = 10; electronic supplementary material, figure S2B).

During behaviour experiments, we recorded animals for 15 min before each experiment to analyse baseline activity and confirm that the arena was fair in the absence of chemosensory cues. Subsequently, 100 µl of a chemical stimulus was gently pipetted into the left side of the arena to minimize mechanosensory disturbances, and larval activity was recorded for another 15 min (figure 1c).

### (b) Video analyses

Video data were obtained and processed as previously described [20] using MULTITRACKER software by Floris van Breugel [44] and

PYTHON v.3.6.2. Additionally, approximate larval length was measured for each animal in IMAGEJ FIJI [45], as the pixel length from head to tail, in a selected video frame that showed the larva in a horizontal position. Lengths were converted to mm using the known inner container width as the conversion ratio. Experimenters were blind to larval sex when measuring lengths. Throughout our analyses, the arena was divided into areas of high concentration ($\geq 50\%$ initial stimulus) and low concentration ($< 50\%$). Larvae could move in a direction that increased local concentration or decreased local concentration. We discounted concentration changes caused by diffusion while the larvae remained immobile. A threshold of $\Delta 2\%\,s^{-1}$ was required to qualify as moving up or down the concentration map.

## (c) Computational modelling

We developed a chemokinetic computational model to investigate larval foraging success in different environments. This model resampled the observed trajectories of *Ae. aegypti* larvae to investigate the consequences of a chemokinetic search strategy using realistic larval behavioural metrics. In the experimental foraging task, simulated animals explored a circular arena until they encountered a food source at the centre of the arena. These arenas included a range of 19 different arena sizes representing many of the ecologically realistic habitats reported in the literature (electronic supplementary material, table S1). The food target was scaled to arena size (comprising 3% of total area) under the assumption that habitats of larger diameter would also contain higher absolute amounts of food. Each simulated larvae began at a random point within the arena, and then explored the environment at each time step by sampling a paired speed-angle data point from experimental data (figure 4*c*(i)). We elected to pair these data points in our model because we observed that the two variables were correlated at higher speeds (figure 4*c*(i)). The time step was re-sampled if the selected data point would cause the trajectory of the simulated larvae to leave the boundary of the experimental arena. Data from animals tested with glucose and amino acids were not included. These experiments were conducted during the manuscript review process, and it was not possible to rerun simulations in the allotted time. Nevertheless, our simulations were sampled from over 700 000 data points from 416 individual larvae. To approximate chemokinetic behaviour, simulated larvae in areas of high food concentration ($> 50\%$) moved slower, and larvae in areas of low food concentration ($\leq 50\%$) moved faster. These differences were implemented by splitting the paired speed-angle data into two bins of equal size, with one bin containing the slowest half of all data points and the other containing the faster half. The probability of sampling from each half was determined as a function of the instantaneous food concentration (figure 4*c*(ii)), with the addition of an exponentially smoothed decision boundary to reduce thresholding artefacts. The empirical data pairs used in these models represented all data taken from larvae observed in clean water before the addition of experimental stimuli, with fed simulations sampling data from fed animals and starved simulations sampling data from starved animals only ($n = 248$ fed, $n = 168$ starved). To define the boundary of 50% food concentration for chemokinetic behavioural decisions, we defined the simulated chemical conditions using an exponential regression model of distance and concentration based on our empirical chemical map (electronic supplementary material, figure S2E). When the simulated larvae entered the food patch at the centre of the arena, the simulation was stopped and the time taken to discover the food was recorded (in seconds). We conducted 1000 simulations for each arena size and nutritional state (fed versus starved).

Data accessibility. Code associated with this manuscript can be found at: https://github.com/eleanorlutz/aedes-aegypti-2019. Data available from the Dryad Digital Repository: https://dx.doi.org/10.5061/dryad.s1rn8pk3n [46].

Authors' contributions. Conceptualization: E.K.L. and J.A.R.; methodology: E.K.L. and J.A.R.; software: E.K.L.; investigation: E.K.L. and T.S.G.; resources: E.K.L. and J.A.R.; data curation: E.K.L.; writing— original draft: E.K.L.; writing—review and editing: E.K.L., J.A.R. and T.S.G.; visualization: E.K.L.; supervision: J.A.R.; project administration: J.A.R.; funding acquisition: E.K.L. and J.A.R.

Competing interests. The authors declare no competing interests.

Funding. This work was supported in part by grants from the National Institute of Health grant no. 1RO1DCO13693 and 1R21AI137947 to J.A.R.; National Science Foundation grant nos IOS-1354159 to J.A.R. and DGE-1256082 to E.K.L.; Air Force Office of Sponsored Research under grant no. FA9550-16-1-0167 to J.A.R.; and the Robin Mariko Harris Award to E.K.L. We thank Floris van Breugel for assistance with video data analysis, the University of Washington Biostatistics Consulting Group for statistical advice, and Binh Nguyen and Kara Kiyokawa for maintaining the Riffell laboratory mosquito colony.

Acknowledgements. We also thank Thomas Daniel, Bingni Brunton, Kameron Harris and the Kincaid 320 Python Club for insightful discussions on programming and data management. Finally, we thank two anonymous reviewers for their contribution of time, expertise and thoughtful advice that significantly improved this manuscript.

## Endnote

[1]Large breeding sites are probably more likely to contain multiple small patches of food distributed throughout the environment, rather than our simulated model of one single patch.

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
