## [Reviewer comments · Proceedings of the Royal Society B: Biological Sciences]

Review History

RSPB-2019-1495.R0 (Original submission)

Review form: Reviewer 1

Recommendation

Accept with minor revision (please list in comments)

Scientific importance: Is the manuscript an original and important contribution to its field?

Acceptable

General interest: Is the paper of sufficient general interest?

Excellent

Quality of the paper: Is the overall quality of the paper suitable?

Good

Is the length of the paper justified?

Yes

Should the paper be seen by a specialist statistical reviewer?

No

Do you have any concerns about statistical analyses in this paper? If so, please specify them explicitly in your report.

No

It is a condition of publication that authors make their supporting data, code and materials available - either as supplementary material or hosted in an external repository. Please rate, if applicable, the supporting data on the following criteria.

Is it accessible?

Yes

Is it clear?

Yes

Is it adequate?

Yes

Do you have any ethical concerns with this paper?

No

Comments to the Author

In their manuscript entitled "Computational and experimental insights into the chemosensory navigation of *Aedes aegypti* mosquito larvae," Lutz, Grewal, and Riffell report a modified behavioral setup to monitor the locomotor activity of single mosquito larvae in response to 6 different chemical stimuli. By analyzing the resulting trajectories, they show that larval behavior is modified by feeding state (fed vs. starved) and identify 2 expected stimuli (food and filtered food extract) as well as a third, yeast RNA extract, that serve as attractive cues in starved, but not fed, larvae. In contrast, quinine is a reliably aversive cue in both fed and starved larvae. The authors suggest that the aggregate responses to three attractive cues (food, food extract, and yeast RNA) are not consistent with chemotaxis or klinokinesis, but are instead indicative of a search strategy called chemokinesis, in which animals do not modulate the direction of their turns but instead non-directionally modulate their speed. Larvae appear to speed up in response to a single aversive cue, and slow down in response to attractive cues, dependent on cue identity and feeding state. A computer model of fed and starved larvae operating with a chemokinetic search strategy informed by the author's data indicate that the modulated behavior of starved larvae improves food discovery in habitats of different size.

The technical quality of the work is excellent and the study is an important addition to an understudied aspect of mosquito biology (larval behavior), but I have concerns about a mismatch between the scope of the experimental data and the broad and definitive conclusions drawn from this data. Further, I feel that the study would be strengthened by examining additional cues and additional concentrations of the cues that are tested.

Major concerns:

1) The number of stimuli tested is low, and does not support the generalization of the study's results to all potentially attractive and aversive cues. For example, fish food (whole or unfiltered) is a complex stimulus that likely contains multiple molecules sensed by multiple receptors and circuits, each eliciting distinct behavioral responses that interact with one another. Further, given the magnitude of the response to food as compared to food extract or yeast RNA, as well as

discrepancies in the change in CS and P between the different stimuli suggests that the three cues are not equivalent, at least at the concentrations tested. For example, whole food has the strongest positive shift in preference, but the smallest effect on CS of the three attractive cues tested, and so I think that the generalization of the findings on search strategy to all putatively attractive and aversive cues is an overreach.

2) I do not find the rationale for testing yeast RNA but not other components of food or food extract to be compelling. The authors should either test other known components of the complex food stimulus (salts, sugars, amino acids, etc.) or provide a stronger rationale for not performing these tests. The review cited on line 121 (ref. 28) does not appear to show data that foraging behavior of *Ae. aegypti* larvae is unaffected by these cues (and certainly not with the sensitivity and precision of the assay described in this study). Furthermore, yeast RNA was the weakest attractive stimulus tested, suggesting that other components of food are important for overall attraction.

3) The rationale for the choice of concentration for each stimulus is unclear. There is a large body of evidence in insect chemotaxis supporting the notion that the same chemical cue can have behavioral effects of opposite valence depending on concentration. Thus, normalizing the arena into 'high' and 'low' concentration areas based on the initial starting concentration may mask the fact that different absolute concentrations of each cue may have important impacts on behavior. The authors test one cue, indole, at two different concentrations, however, indole does not have strong behavioral effects at either concentration studied. The study would be significantly strengthened by the authors testing at minimum two additional concentrations (one higher, one lower) of each behaviorally-relevant cue to see if the principal conclusions of the work hold.

Minor concerns:

4) The authors should present the data as in Figure 1D for the other 4 odorants and for both starved and fed larvae as a supplemental figure. Those visualizations are informative and intuitive to interpret.

5) In Figure 3 and S3 it would be helpful to label each panel with the full name of the parameter being analyzed (i.e. label Figure 3C with 'Stimulus preference.')

6) One prediction of a chemokinetic model of behavior is that the effect on locomotor parameters should not depend on the presence of a gradient, as evidenced by the effects seen on CS but not DS in the present study. In other words, one might expect that the addition of a uniform concentration of quinine would cause an increase in locomotor speed. The authors should consider doing these experiments, if feasible.

Typographical errors:

Line 128: the authors refer to seven stimuli (perhaps counting each concentration of indole?) where in line 96 and elsewhere they refer to six cues. These should be standardized.

Review form: Reviewer 2

Recommendation

Accept with minor revision (please list in comments)

Scientific importance: Is the manuscript an original and important contribution to its field?

Excellent

General interest: Is the paper of sufficient general interest?

Good

Quality of the paper: Is the overall quality of the paper suitable?

Excellent

Is the length of the paper justified?

Yes

Should the paper be seen by a specialist statistical reviewer?

No

Do you have any concerns about statistical analyses in this paper? If so, please specify them explicitly in your report.

Yes

It is a condition of publication that authors make their supporting data, code and materials available - either as supplementary material or hosted in an external repository. Please rate, if applicable, the supporting data on the following criteria.

Is it accessible?

Yes

Is it clear?

Yes

Is it adequate?

Yes

Do you have any ethical concerns with this paper?

No

Comments to the Author

The manuscript by Lutz et al analyses a very relevant aspect of mosquito behavior that to date has received poor attention. Authors have elegantly shown that larvae use chemical cues (not signals like they eventually mention) in order to forage in the small containers where they frequently breed. Experiments seem mostly well designed and approaches used to model larval behavior seem fundamentally correct. The text is nicely written and results are properly discussed in the frame of questions that are relevant to understand the ecology and evolution of this very relevant urban disease vector. As this is a very good example of experimental study of larval behavior (few exist) I foresee impact as a founding work and interest by a large audience including vector biologists, chemical ecologists, theoretical scholars dedicated to study behavior and those focused on mathematical modeling of biological processes.

I have several minor comments listed below that deserve attention in order to improve the manuscript.

Introduction

line 45, I suggest eliminating "the" from the phrase, as these are by no means all the possible cues that deserve to be evaluated.

line 57, please check syntax (our results-highlights).

Results

- At the start of this section it would be desirable to make clear that authors studied the behavior of 4th instar larvae exclusively. I assume this, as it can be inferred from the supplemental materials and methods. I emphasize that this is fine to have focused on 4th instar larvae, as previous instars may not be tractable by current video methods. Nevertheless, I consider that a statement in this sense would let the reader get a proper impression about the experimental model used.

- lines 90-92, the use of the term "signals" does not seem proper here because their study focuses on "cues" related to the foraging context.

- lines 93-96, according to this phrase authors consider food as one of the six chemosensory cues. Nevertheless, food is not only a complex set of chemosensory stimuli (I not sure it would be correct to call that set a cue), but one including mechanosensory information. As such, I would not consider it in the general description made in this phrase, but as an initial positive control for foraging behavior. To be strict, not even the extract is a single chemosensory cue, but probably a very complex set of them.

- lines 96-99, where is this information depicted? Please refer to a result (either figure or table) that can be seen by the reader, not only the outcome of a statistical test.

- lines 124-126, authors mention seven stimuli, but as far as I could check, they have only listed six. I understand the seventh is water, but it seems to this reviewer that it was not mentioned until this point (first mentioned 12 lines below).

-lines 154-156, again, authors group all stimuli tested separately as in the "chemosensory" category. This seems simply incorrect and may be replaced by a broader expression, e.g., "seven sets of stimuli".

- lines 209-212, can authors estimate what does "high" mean in this study? If possible, it would be informative because the phrase seems to suggest that larvae slow down their swimming above a threshold concentration of attractant.

- lines 237-241, authors limited maximum container size to 20cm based on an old literature report (Table S1). Nevertheless, several subsequent studies have shown that breeding site productivity is clearly related to container size. Furthermore, larger containers like water tanks and metal drums are considered key targets for controlling *Ae. aegypti* due to their high larval productivity. I consider that it would be very enriching to see much larger sizes included in the modelling, if possible.

Discussion

- any effects of larval sex on behavior are not mentioned in this or the previous section. This reviewer is not sure if none were observed, but even that does not seem to be reported.

- lines 283-285, this phrase does not seem totally logical to me. Why do authors make a connection between ATP perception during adult feeding and RNA detection by larvae?

- lines 289-293, this phrase needs to be supported by a literature reference to the work with *Drosophila melanogaster*. Besides, it seems to have a syntax problem ("A gustatory or ionotropic receptor" and "candidates" (I acknowledge I am not a native English speaker and therefore, my perspective may be wrong).

- lines 293-296, please check, as reference 22 does not seem to mention testing quinine.

Material and methods

- lines 391-393, this phrase is not correct and it needs to be modified. Natural breeding sites are very frequently larger than 20cm in diameter, reaching up to 100cm or more. For example, authors can refer to Maciel-de-Freitas, R, et al. *Memórias do Instituto Oswaldo Cruz* 102.4 (2007): 489-496. Other studies shown larger breeding sites as more productive too.

- lines 406-410, why if authors indicate that no klinokinetic behavior was observed, modelling includes changes in turning rates between high and low food concentration areas?

Finally, in the supplemental section authors do not tell why they used the non-parametric Mann Whitney or Kruskal Wallis tests. How was this decided?

Decision letter (RSPB-2019-1495.R0)

29-Jul-2019

Dear Dr Riffell:

Your manuscript has now been peer reviewed and the reviews have been assessed by an Associate Editor. The reviewers' comments (not including confidential comments to the Editor) and the comments from the Associate Editor are included at the end of this email for your reference. As you will see, the reviewers and the Editors have raised some concerns with your manuscript and we would like to invite you to revise your manuscript to address them.

Research ethics:

Use of animals and field studies:

It is a condition of publication that you make available the data and research materials supporting the results in the article. Datasets should be deposited in an appropriate publicly available repository and details of the associated accession number, link or DOI to the datasets

must be included in the Data Accessibility section of the article (<https://royalsociety.org/journals/ethics-policies/data-sharing-mining/>). Reference(s) to datasets should also be included in the reference list of the article with DOIs (where available).

If you wish to submit your data to Dryad (<http://datadryad.org/>) and have not already done so you can submit your data via this link [http://datadryad.org/submit?journalID=RSPB&manu=\(Document not available\)](http://datadryad.org/submit?journalID=RSPB&manu=(Document%20not%20available)), which will take you to your unique entry in the Dryad repository.

Please submit a copy of your revised paper within three weeks. If we do not hear from you within this time your manuscript will be rejected. If you are unable to meet this deadline please let us know as soon as possible, as we may be able to grant a short extension.

Best wishes,
Professor Gary Carvalho
mailto: proceedingsb@royalsociety.org

Associate Editor
Comments to Author:
Associate Editor: Doug Altshuler

Lutz et al. have conducted a very interesting, integrated study of chemosensory navigation in larvae of *Aedes aegypti*. This mosquito is obviously a critical disease vector so any new insight into the mechanisms of its growth are potentially important. Through a series of experiments and then some modeling, the authors identify a search strategy of chemokinesis. The referees found that experiments and modeling supporting this strategy to be convincing, and I agree. Referee 1 pointed out that some of the treatments were modest relative to the language describing the claims. I would encourage that point be addressed through tempering some of the language, but

they could also consider additional treatments if this appeals. Referee 2 also provides a number of suggestions that will be helpful should the authors elect to revise the manuscript.

Reviewer(s)' Comments to Author:

Referee: 1

Comments to the Author(s)

In their manuscript entitled "Computational and experimental insights into the chemosensory navigation of *Aedes aegypti* mosquito larvae," Lutz, Grewal, and Riffell report a modified behavioral setup to monitor the locomotor activity of single mosquito larvae in response to 6 different chemical stimuli. By analyzing the resulting trajectories, they show that larval behavior is modified by feeding state (fed vs. starved) and identify 2 expected stimuli (food and filtered food extract) as well as a third, yeast RNA extract, that serve as attractive cues in starved, but not fed, larvae. In contrast, quinine is a reliably aversive cue in both fed and starved larvae. The authors suggest that the aggregate responses to three attractive cues (food, food extract, and yeast RNA) are not consistent with chemotaxis or klinokinesis, but are instead indicative of a search strategy called chemokinesis, in which animals do not modulate the direction of their turns but instead non-directionally modulate their speed. Larvae appear to speed up in response to a single aversive cue, and slow down in response to attractive cues, dependent on cue identity and feeding state. A computer model of fed and starved larvae operating with a chemokinetic search strategy informed by the author's data indicate that the modulated behavior of starved larvae improves food discovery in habitats of different size.

The technical quality of the work is excellent and the study is an important addition to an understudied aspect of mosquito biology (larval behavior), but I have concerns about a mismatch between the scope of the experimental data and the broad and definitive conclusions drawn from this data. Further, I feel that the study would be strengthened by examining additional cues and additional concentrations of the cues that are tested.

Major concerns:

1) The number of stimuli tested is low, and does not support the generalization of the study's results to all potentially attractive and aversive cues. For example, fish food (whole or unfiltered) is a complex stimulus that likely contains multiple molecules sensed by multiple receptors and circuits, each eliciting distinct behavioral responses that interact with one another. Further, given the magnitude of the response to food as compared to food extract or yeast RNA, as well as discrepancies in the change in CS and P between the different stimuli suggests that the three cues are not equivalent, at least at the concentrations tested. For example, whole food has the strongest positive shift in preference, but the smallest effect on CS of the three attractive cues tested, and so I think that the generalization of the findings on search strategy to all putatively attractive and aversive cues is an overreach.

2) I do not find the rationale for testing yeast RNA but not other components of food or food extract to be compelling. The authors should either test other known components of the complex food stimulus (salts, sugars, amino acids, etc.) or provide a stronger rationale for not performing these tests. The review cited on line 121 (ref. 28) does not appear to show data that foraging behavior of *Ae. aegypti* larvae is unaffected by these cues (and certainly not with the sensitivity and precision of the assay described in this study). Furthermore, yeast RNA was the weakest attractive stimulus tested, suggesting that other components of food are important for overall attraction.

3) The rationale for the choice of concentration for each stimulus is unclear. There is a large body of evidence in insect chemotaxis supporting the notion that the same chemical cue can have behavioral effects of opposite valence depending on concentration. Thus, normalizing the arena into 'high' and 'low' concentration areas based on the initial starting concentration may mask the fact that different absolute concentrations of each cue may have important impacts on behavior. The authors test one cue, indole, at two different concentrations, however, indole does not have strong behavioral effects at either concentration studied. The study would be significantly strengthening by the authors testing at minimum two additional concentrations (one higher, one lower) of each behaviorally-relevant cue to see if the principal conclusions of the work hold.

Minor concerns:

4) The authors should present the data as in Figure 1D for the other 4 odorants and for both starved and fed larvae as a supplemental figure. Those visualizations are informative and intuitive to interpret.

5) In Figure 3 and S3 it would be helpful to label each panel with the full name of the parameter being analyzed (i.e. label Figure 3C with 'Stimulus preference.')

6) One prediction of a chemokinetic model of behavior is that the effect on locomotor parameters should not depend on the presence of a gradient, as evidenced by the effects seen on CS but not DS in the present study. In other words, one might expect that the addition of a uniform concentration of quinine would cause an increase in locomotor speed. The authors should consider doing these experiments, if feasible.

Typographical errors:

Line 128: the authors refer to seven stimuli (perhaps counting each concentration of indole?) where in line 96 and elsewhere they refer to six cues. These should be standardized.

Referee: 2

Comments to the Author(s)

The manuscript by Lutz et al analyses a very relevant aspect of mosquito behavior that to date has received poor attention. Authors have elegantly shown that larvae use chemical cues (not signals like they eventually mention) in order to forage in the small containers where they frequently breed. Experiments seem mostly well designed and approaches used to model larval behavior seem fundamentally correct. The text is nicely written and results are properly discussed in the frame of questions that are relevant to understand the ecology and evolution of this very relevant urban disease vector. As this is a very good example of experimental study of larval behavior (few exist) I foresee impact as a founding work and interest by a large audience including vector biologists, chemical ecologists, theoretical scholars dedicated to study behavior and those focused on mathematical modeling of biological processes.

I have several minor comments listed below that deserve attention in order to improve the manuscript.

Introduction

line 45, I suggest eliminating "the" from the phrase, as these are by no means all the possible cues that deserve to be evaluated.

line 57, please check syntax (our results-highlights).

Results

- At the start of this section it would be desirable to make clear that authors studied the behavior of 4th instar larvae exclusively. I assume this, as it can be inferred from the supplemental materials and methods. I emphasize that this is fine to have focused on 4th instar larvae, as previous instars may not be tractable by current video methods. Nevertheless, I consider that a statement in this sense would let the reader get a proper impression about the experimental model used.
- lines 90-92, the use of the term "signals" does not seem proper here because their study focuses on "cues" related to the foraging context.
- lines 93-96, according to this phrase authors consider food as one of the six chemosensory cues. Nevertheless, food is not only a complex set of chemosensory stimuli (I not sure it would be correct to call that set a cue), but one including mechanosensory information. As such, I would not consider it in the general description made in this phrase, but as an initial positive control for foraging behavior. To be strict, not even the extract is a single chemosensory cue, but probably a very complex set of them.
- lines 96-99, where is this information depicted? Please refer to a result (either figure or table) that can be seen by the reader, not only the outcome of a statistical test.
- lines 124-126, authors mention seven stimuli, but as far as I could check, they have only listed six. I understand the seventh is water, but it seems to this reviewer that it was not mentioned until this point (first mentioned 12 lines below).
- lines 154-156, again, authors group all stimuli tested separately as in the "chemosensory" category. This seems simply incorrect and may be replaced by a broader expression, e.g., "seven sets of stimuli".
- lines 209-212, can authors estimate what does "high" mean in this study? If possible, it would be informative because the phrase seems to suggest that larvae slow down their swimming above a threshold concentration of attractant.
- lines 237-241, authors limited maximum container size to 20cm based on an old literature report (Table S1). Nevertheless, several subsequent studies have shown that breeding site productivity is clearly related to container size. Furthermore, larger containers like water tanks and metal drums are considered key targets for controlling *Ae. aegypti* due to their high larval productivity. I consider that it would be very enriching to see much larger sizes included in the modelling, if possible.

Discussion

- any effects of larval sex on behavior are not mentioned in this or the previous section. This reviewer is not sure if none were observed, but even that does not seem to be reported.
- lines 283-285, this phrase does not seem totally logical to me. Why do authors make a connection between ATP perception during adult feeding and RNA detection by larvae?
- lines 289-293, this phrase needs to be supported by a literature reference to the work with *Drosophila melanogaster*. Besides, it seems to have a syntax problem ("A gustatory or ionotropic receptor" and "candidates" (I acknowledge I am not a native English speaker and therefore, my perspective may be wrong).
- lines 293-296, please check, as reference 22 does not seem to mention testing quinine.

Material and methods

- lines 391-393, this phrase is not correct and it needs to be modified. Natural breeding sites are very frequently larger than 20cm in diameter, reaching up to 100cm or more. For example, authors can refer to Maciel-de-Freitas, R, et al. *Memórias do Instituto Oswaldo Cruz* 102.4 (2007): 489-496. Other studies shown larger breeding sites as more productive too.
- lines 406-410, why if authors indicate that no klinokinetic behavior was observed, modelling includes changes in turning rates between high and low food concentration areas?

Finally, in the supplemental section authors do not tell why they used the non-parametric Mann Whitney or Kruskal Wallis tests. How was this decided?

Author's Response to Decision Letter for (RSPB-2019-1495.R0)

See Appendix A.

Decision letter (RSPB-2019-1495.R1)

24-Oct-2019

Dear Dr Riffell

I am pleased to inform you that your manuscript entitled "Computational and experimental insights into the chemosensory navigation of *Aedes aegypti* mosquito larvae" has been accepted for publication in Proceedings B.

Open Access

Paper charges

Sincerely,

Professor Gary Carvalho
Editor, Proceedings B
mailto: proceedingsb@royalsociety.org

Associate Editor:
Comments to Author:
Associate Editor: Doug Altshuler

The authors have done an excellent job in addressing the reviewers' concerns. This is a strong contribution that adds significantly to our understanding of mosquito behavior and its mechanistic basis.

Appendix A

We thank the editor and reviewer for the helpful comments. Based on those comments, we have:

- Conducted additional experiments with glucose and a mixture of amino acids to examine whether these food-related cues may evoke behavioral responses.
- Added detail on the range of concentrations tested.
- Conducted additional modeling experiments to examine larger arena sizes.

In addition, wherever possible we made changes to the manuscript based on the reviewer and editor comments. Those changes are detailed in the manuscript (red text) and described below.

Editor Comments:

Lutz et al. have conducted a very interesting, integrated study of chemosensory navigation in larvae of *Aedes aegypti*. This mosquito is obviously a critical disease vector so any new insight into the mechanisms of its growth are potentially important. Through a series of experiments and then some modeling, the authors identify a search strategy of chemokinesis. The referees found that experiments and modeling supporting this strategy to be convincing, and I agree. Referee 1 pointed out that some of the treatments were modest relative to the language describing the claims. I would encourage that point be addressed through tempering some of the language, but they could also consider additional treatments if this appeals. Referee 2 also provides a number of suggestions that will be helpful should the authors elect to revise the manuscript.

AUTHOR RESPONSE: In response to Reviewer 1's concerns, we conducted experiments on 83 additional larvae. Please note that we have recalculated several statistical tests to include these additional animals. In the manuscript we have highlighted these statistical test results in blue; changes to the text are in red. Our overall conclusions did not change based on these new experiments. However, some statistical values changed slightly due to the addition of more treatments (affecting the Bonferroni-Holm correction) or more animals (affecting comparisons between fed / starved and male / female animals).

Reviewer 1:

Major concerns:

1) The number of stimuli tested is low, and does not support the generalization of the study's results to all potentially attractive and aversive cues. For example, fish food (whole or unfiltered) is a complex stimulus that likely contains multiple molecules sensed by multiple receptors and circuits, each eliciting distinct behavioral responses that interact with one another. Further, given the magnitude of the response to food as compared to food extract or yeast RNA, as well as discrepancies in the change in CS and P between the different stimuli suggests that the three cues are not equivalent, at least at the concentrations tested. For example, whole food has the strongest positive shift in preference, but the smallest effect on CS of the three attractive cues tested, and so I think that the generalization of the findings on a search strategy to all putatively attractive and aversive cues is an overreach.

AUTHOR RESPONSE: We thank the reviewer for the helpful comments. We did not intend to convey that all possible cues may be equivalent, and we have revised the language of our manuscript to better convey our conclusions. In particular, we have changed the section heading for these results from "Larvae use chemokinesis to navigate to chemical cues" to "A chemokinesis navigation strategy is most consistent with larval aggregation toward cues investigated in this study." We have also conducted 83 additional experiments testing two more cues.

Although there do appear to be small differences in CS between attractive cues, our CS measurement was designed to be used only in addition with DS, DP, etc. to identify an overall navigation strategy. To further compare the speed relationships between stimuli, we believe a different measurement would be more appropriate. The reason is that CS compares larval activity in areas of high concentration to activity in areas of low concentration. For highly attractive cues like food, larvae spend very little time in areas of low concentration, and CS measurements thus contain more noise than for stimuli where animals spend equal amounts of time in high and low concentration areas.

Although this CS measurement is sufficient to identify an overall navigation strategy, it is not a specific enough tool to answer questions about relative stimulus preference. For this reason, in this manuscript we use only the preference variable P - which is not affected by this noise issue - to assess animal preference towards different odors.

2) I do not find the rationale for testing yeast RNA but not other components of food or food extract to be compelling. The authors should either test other known components of the complex food stimulus (salts, sugars, amino acids, etc.) or provide a stronger rationale for not performing these tests. The review cited on line 121 (ref. 28) does not appear to show data that foraging behavior of *Ae. aegypti* larvae is unaffected by these cues (and certainly not with the sensitivity and precision of the assay described in this study). Furthermore, yeast RNA was the weakest attractive stimulus tested, suggesting that other components of food are important for overall attraction.

AUTHOR RESPONSE: We have conducted additional experiments on 83 animals testing glucose and a mixture of amino acids - two more major components of larval food. This amino acid mixture includes nine essential amino acids required for *Ae. aegypti* larval development, in concentrations previously shown to support optimal larval growth in laboratory conditions (Singh and Brown 1957). We have also made an effort to describe the limitations of our stimulus panel in the Discussion section.

3) The rationale for the choice of concentration for each stimulus is unclear. There is a large body of evidence in insect chemotaxis supporting the notion that the same chemical cue can have behavioral effects of opposite valence depending on concentration. Thus, normalizing the arena into 'high' and 'low' concentration areas based on the initial starting concentration may mask the fact that different absolute concentrations of each cue may have important impacts on behavior. The authors test one cue, indole, at two different concentrations, however, indole does not have strong behavioral effects at either concentration studied. The study would be significantly strengthening by the authors testing at minimum two additional concentrations (one higher, one lower) of each behaviorally-relevant cue to see if the principal conclusions of the work hold.

AUTHOR RESPONSE: We have revised the manuscript to better explain our rationale for choice of concentrations (Electronic Materials and Methods section). The initial concentration used in this study (100uM) was selected based on a previous study by Zwiebel et al. 2008, which showed that *An. gambiae* mosquito larvae significantly prefer indole and o-cresol at 100uM. When we observed no responses by *Ae. aegypti* to 100uM indole, we next tested 10mM indole - shown by Zwiebel et al. to have a significant repellent effect in *An. gambiae* - and still observed no behavioral response in *Ae. aegypti*. Thus, we believe our null results are interesting in the context of responses observed in other larval mosquito species.

Additionally, we mapped the distribution of stimuli within our arena (Figure S2E), and thus are confident that our experiments using 10mM and 100uM indole spanned the full range of concentrations to trace levels, including concentrations much lower than 100uM. Because indole is not soluble in water above ~16mM, our experiments at 10mM are already near the limit of the highest possible concentration.

Nevertheless, we understand the concern that our interpretation of “high” and “low” concentrations might mask larval preference for intermediate concentrations. To visualize the full range of larval responses, we have added three new supplemental figures (S3, S4, S5). These figures visualize the distribution of larvae throughout the entire 15 minute experiment. We did not observe any peaks in larval distribution that would suggest preference for an intermediate concentration - larvae aggregated either at the highest concentration (food, food extract, yeast), the lowest concentration (aversive cues like quinine), or at each wall (fed animals, which aggregate near walls regardless of stimulus type).

Minor concerns:

4) The authors should present the data as in Figure 1D for the other 4 odorants and for both starved and fed larvae as a supplemental figure. Those visualizations are informative and intuitive to interpret.

AUTHOR RESPONSE: We have added this data visualization for all experimental stimuli in two new Supplemental Figures (S3 and S4).

5) In Figure 3 and S3 it would be helpful to label each panel with the full name of the parameter being analyzed (i.e. label Figure 3C with ‘Stimulus preference.’)

AUTHOR RESPONSE: We have revised Figure 3 and Figure S6 (formerly Figure S3) with labeled parameters.

6) One prediction of a chemokinetic model of behavior is that the effect on locomotor parameters should not depend on the presence of a gradient, as evidenced by the effects seen on CS but not DS in the present study. In other words, one might expect that the addition of a uniform concentration of quinine would cause an increase in locomotor speed. The authors should consider doing these experiments, if feasible.

AUTHOR RESPONSE: We thank the reviewer for the suggestion of an interesting experiment. However, the suggested experiment would require entirely different experimental and analysis methods, and therefore may be better suited for a future research study. There are several reasons why the suggested experiment is not feasible with our current methods. To create a uniform concentration of quinine in our current arena, it would be necessary to manually mix the water after addition - which would likely introduce significant behavioral artifacts due to mechanical disturbance. If instead we added the larva directly to quinine, we would not be able to acclimate the larvae to the experimental arena, or quantify its behavior before the experiment (a necessary step for all data analyses conducted in this paper). Finally, without any areas of low quinine, it may be difficult to determine whether larval responses are affected by physiological confounds such as fatigue, toxic effects of quinine, or receptor adaptation.

Reviewer 2:

Introduction

line 45, I suggest eliminating "the" from the phrase, as these are by no means all the possible cues that deserve to be evaluated.

line 57, please check syntax (our results-highlights).

AUTHOR RESPONSE: We thank the reviewer for their helpful comments, and have revised the introduction with their suggestions.

Results

- At the start of this section it would be desirable to make clear that authors studied the behavior of 4th instar larvae exclusively. I assume this, as it can be inferred from the supplemental materials and methods. I emphasize that this is fine to have focused on 4th instar larvae, as previous instars may not be tractable by current video methods. Nevertheless, I consider that a statement in this sense would let the reader get a proper impression about the experimental model used.

AUTHOR RESPONSE: We have added this information at the beginning of the Results section.

- lines 90-92, the use of the term "signals" does not seem proper here because their study focuses on "cues" related to the foraging context.

AUTHOR RESPONSE: We have replaced the term "signal" with "cue".

- lines 93-96, according to this phrase authors consider food as one of the six chemosensory cues. Nevertheless, food is not only a complex set of chemosensory stimuli (I not sure it would be correct to call that set a cue), **but one including mechanosensory information.** As such, I would not consider it in the general description made in this phrase, but as an initial positive control for foraging behavior. To be strict, not even the extract is a single chemosensory cue, but probably a very complex set of them.

AUTHOR RESPONSE: We have revised our description of these experimental treatments as "stimuli" rather than "chemosensory cues" throughout the manuscript to the best of our ability.

- lines 96-99, where is this information depicted? Please refer to a result (either figure or table) that can be seen by the reader, not only the outcome of a statistical test.

AUTHOR RESPONSE: We have revised Supplemental Figure 2 to include this information (panel F), and cited this figure in lines 96-99.

- lines 124-126, authors mention seven stimuli, but as far as I could check, they have only listed six. I understand the seventh is water, but it seems to this reviewer that it was not mentioned until this point (first mentioned 12 lines below).

AUTHOR RESPONSE: We have standardized the description of "stimuli" in this manuscript to exclude water. We have also added two additional stimuli during our revision process, so we now describe eight "stimuli" in addition to water (indole, o-cresol, quinine, food, food extract, yeast RNA, glucose, and an amino acid mixture).

-lines 154-156, again, authors group all stimuli tested separately as in the "chemosensory" category. This seems simply incorrect and may be replaced by a broader expression, e.g., "seven sets of stimuli".

AUTHOR RESPONSE: We have fixed this description, and attempted to replace the term "chemosensory cue" throughout the paper with "stimulus."

- lines 209-212, can authors estimate what does "high" mean in this study? If possible, it would be informative because the phrase seems to suggest that larvae slow down their swimming above a threshold concentration of attractant.

AUTHOR RESPONSE: Our concentration map can give a general estimate of "high" and "low" concentrations as detected by the larvae, and we have created a new supplemental figure (Figure S5) to visualize this information. This visualization depicts the larval preference normalized to the acclimation period, binned by concentration.

- lines 237-241, authors limited maximum container size to 20cm based on an old literature report (Table S1). Nevertheless, several subsequent studies have shown that breeding site productivity is clearly related to container size. Furthermore, larger containers like water tanks and metal drums are considered key targets for controlling *Ae. aegypti* due to their high larval productivity. I consider that it would be very enriching to see much larger sizes included in the modelling, if possible.

AUTHOR RESPONSE: We have added three additional container sizes to the modeling experiments, and included these results in Figure 4E as well as in the Results section.

Discussion

- any effects of larval sex on behavior are not mentioned in this or the previous section. This reviewer is not sure if none were observed, but even that does not seem to be reported.

AUTHOR RESPONSE: In Figure S1 we compared sex-specific physiological features of larvae by pooling measurements of all animals from the acclimation period. However, our experiment results for each individual stimulus have far fewer animals for each sex, and we did not have sufficient power to analyze stimulus-specific sex differences. Nevertheless, we accounted for possible sex-specific confounds such as larval size or movement speed by normalizing the stimulus response of each animal to its activity during the pre-experiment acclimation period.

To better inform readers, we have added this information to the figure caption to Figure S1. In addition, we have included the sex information for each animal in our open-source code and data. We hope that the availability of this data can help inform researchers developing future experiments, even if statistical comparisons cannot be drawn with our current sample size.

- lines 283-285, this phrase does not seem totally logical to me. Why do authors make a connection between ATP perception during adult feeding and RNA detection by larvae?

AUTHOR RESPONSE: We have attempted to clarify this point by rephrasing the sentence to read: "Although adult *Ae. aegypti* feeding is regulated by ATP perception, we are unaware of other work demonstrating perception of nucleotides or nucleic acids such as RNA in *Ae. aegypti* larvae."

- lines 289-293, this phrase needs to be supported by a literature reference to the work with *Drosophila melanogaster*. Besides, it seems to have a syntax problem ("A gustatory or ionotropic receptor" and "candidates" (I acknowledge I am not a native English speaker and therefore, my perspective may be wrong)).

AUTHOR RESPONSE: We have removed this phrase, as after looking closely at our original sources we have realized that this statement is based on unpublished work.

- lines 293-296, please check, as reference 22 does not seem to mention testing quinine.

AUTHOR RESPONSE: We have removed this phrase.

Material and methods

- lines 391-393, this phrase is not correct and it needs to be modified. Natural breeding sites are very frequently larger than 20cm in diameter, reaching up to 100cm or more. For example, authors can refer to Maciel-de-Freitas, R, et al. Memórias do Instituto Oswaldo Cruz 102.4 (2007): 489-496. Other studies shown larger breeding sites as more productive too.

AUTHOR RESPONSE: We have corrected the phrasing of breeding site sizes throughout the manuscript to address larger sites. We have also conducted more simulations to include three larger breeding sites (Figure 4E and Results section).

- lines 406-410, why if authors indicate that no klinokinetic behavior was observed, modelling includes changes in turning rates between high and low food concentration areas?

AUTHOR RESPONSE: The modeling did not specifically include changes in turning rate, and we have fixed this mistake in the description. We apologize for our error in proofreading the manuscript before submission.

Finally, in the supplemental section authors do not tell why they used the non-parametric Mann Whitney or Kruskal Wallis tests. How was this decided?

AUTHOR RESPONSE: We thank the reviewer for pointing out this omission! A non-parametric Mann Whitney test was used to compare larval size, because a Shapiro-Wilk normality test determined that these datasets were not normally distributed. Similarly, we used the Kruskal Wallis test to compare larval behavioral metrics, because the measurement variables were not normally distributed. We have added this explanation into the “Statistical Analysis” supplemental section.